# Mapping near real-time soil moisture dynamics over Tasmania with transfer learning

Marliana Tri Widyastuti[1], José Padarian[1], Budiman Minasny[1], Mathew Webb[2], Muh Taufik[3], Darren Kidd[2]

[1]School of Life an Environmental Sciences, The University of Sydney, Sydney, New South Wales, Australia.
[2]Environment, Heritage & Land Division, Department of Natural Resources and Environment Tasmania, Prospect, Tasmania, Australia
[3]Department of Geophysics and Meteorology, IPB University, Jalan Meranti Wing 19 Level 4 Darmaga Campus, Bogor, Indonesia 16680

*Correspondence to*: Marliana T. Widyastuti (marlianatri.widyastuti@sydney.edu.au)

**Abstract.** Soil moisture, an essential parameter for hydroclimatic studies, exhibits significant spatial and temporal variability, making it challenging to map at fine spatiotemporal resolutions. Although current remote sensing products provide global soil moisture estimate at a fine temporal resolution, they are at a coarse spatial resolution. In recent years, deep learning (DL) has been applied to generate high-resolution maps of various soil properties, but DL requires a large amount of training data. This study aimed to map daily soil moisture across Tasmania, Australia at 80 meters resolution based on a limited set of training data. We assessed three modelling strategies: DL models calibrated using an Australian dataset (51,411 observation points), models calibrated using the Tasmanian dataset (9,825 observation points), and a transfer learning technique that transferred information from the Australian models to Tasmania. We also evaluated two DL approaches, i.e. Multilayer perceptron (MLP) and Long Short-Term Memory (LSTM). The models included Soil Moisture Active Passive (SMAP) dataset, weather data, elevation map, land cover and multilevel soil properties maps as inputs to generate soil moisture at the surface (0-30 cm) and subsurface (30-60 cm) layers. Results showed that (1) models calibrated from the Australian dataset performed worse than Tasmanian models regardless of the type of DL approaches; (2) Tasmanian models, calibrated solely using local data, resulted in shortcomings in predicting soil moisture; and (3) Transfer learning exhibited remarkable performance improvements (error reductions of up to 45% and a 50% increase in correlation) and resolved the drawbacks of the previous two models. The LSTM models with transfer learning had the highest overall performance with an average mean absolute error (MAE) of 0.07 $m^3m^{-3}$ and a correlation coefficient (r) of 0.77 across stations for surface layer and MAE = 0.07 $m^3m^{-3}$, and r = 0.69 for subsurface layer. The fine-resolution soil moisture maps captured the detailed landscape variation as well as temporal variation according to four distinct seasons in Tasmania. The models were then applied to generate daily soil moisture maps of Tasmania, integrated into a near-real-time monitoring system to assist agricultural decision making.

## 1 Introduction

Soil moisture (SM) plays an essential role in land modelling as it links the natural system's components of soil, climate, and plants. In hydrology, this variable is commonly used as a proxy to assess hydrological extreme events such as drought

assessment (Taufik et al., 2022; Lin et al., 2023). In agricultural practices, soil moisture can provide valuable information for soil-water management and crop yield predictions (Yang et al., 2021). Measuring and mapping soil moisture has challenged
soil scientists as it is highly spatially and temporally diverse. Soil moisture variation is characterised by climate zone, topographic features, vegetation cover, and soil characteristics, including clay content, soil aggregation, and organic carbon content (Minasny and Mcbratney, 2003; Védère et al., 2022).

Globally, soil moisture information is available in various formats and coverage. At the point scale, the International Soil Moisture Network provides a harmonised measured soil moisture database worldwide (Dorigo et al., 2021). In Australia, soil
moisture observations can be found in the Oz-Net and Oz-Flux databases (Smith et al., 2012; Beringer et al., 2016). However, despite accurate information on observations at the point level, the spatial coverage of these measurements is limited, meaning that soil moisture content in unmonitored areas is uncertain. To bridge this gap, various spatial datasets were generated to complement the point scale measurements. Remote sensing, geostatistical models, water balance models, or a combination of them are the principal methods to derive SM images that cover space and time.

Notable soil moisture across Australia continent include the Australian Water Resource Assessment Landscape (AWRA-L), which provides a 5-km resolution soil moisture level based on the water balance approach (Frost et al., 2016). Using the OzFlux and OzNet data points as input, this dataset covers moisture prediction at three soil layers (0-10 cm, 10-100 cm, and 100-600 cm). The Soil Moisture Integration and Prediction System (SMIPS) provides daily soil water balance map at 1 km resolution by integrating machine learning and water balance models (Wimalathunge and Bishop, 2019; Stenson et al., 2021). This
product represents proportion of available water within the 90 cm soil layer and is updated daily with a latency of 3 days.

In addition, several global spatial datasets are available as near-present soil moisture maps at various spatial and temporal resolutions. The Global Land Data Assimilation System (GLDAS) products offer the estimated soil moisture from Noah model at surface (0-2 cm) and rootzone (0-100cm) layers (Li et al., 2019). The GLDAS images are spatially at 0.25 to 1 degree with 3-hour to daily temporal resolution, updated daily with 1 month latency. The ERA5-Land provides four levels of daily soil
moisture (0-7, 7-28, 28-100, 100-289 cm depth) at 0.1-degree spatial resolution with a 2- to 3-month delay (Muñoz-Sabater et al., 2021). Soil Moisture Active Passive level 4 (SMAP-L4), as the most recent product, provides vertical average of soil moisture at surface (0-5 cm) and rootzone (0-100 cm) layers based on NASA's Catchment land surface model assimilated with SMAP L-band (Reichle et al., 2017). With its shortest latency time, the SMAP product is widely used in continuous monitoring systems. SMAP data requires downscaling for higher spatial resolution, enhancing its reliability for agricultural and
environmental monitoring. Several studies have addressed this by developing finer-resolution maps (Cai et al., 2022; Hu et al., 2020; Wei et al., 2019; Xu et al., 2022; Xu et al., 2021; Li et al., 2022b; Dashtian et al., 2024).

Recent advances in deep learning (DL) have enabled the production of high-resolution maps of soil properties in recent years (Padarian et al., 2020; Padarian et al., 2019b; Behrens et al., 2018; Minasny et al., 2024). DL algorithms have been assessed to map soil moisture at high spatial resolution (Fuentes et al., 2022). Additionally, several studies using DL models have been
investigated to downscale the global soil moisture dataset based on point data observations. However, most studies attempted

to produce 1 km resolution maps, which are still too coarse for agricultural management as they cannot capture the highly variable soil and topography (Zhao et al., 2022; Cai et al., 2022; Alemohammad et al., 2018; Li et al., 2022c).

Despite its high applicability, the performance of DL models is highly influenced by the amount of data for model development (Gütter et al., 2022; Ng et al., 2020). Small datasets may lead to overfitting during the model training and can further impact the final model accuracy (Ng et al., 2020). To address the issue of a small training dataset, several studies employed transfer learning (TL) techniques, to leverage models created from a larger dataset. TL works by transferring the information derived from a model trained from a large dataset to a new model with a similar architecture. This technique is commonly used to increase the performance of models built from a limited number of observations (Yao et al., 2023). Several studies, particularly in soil science, have implemented this technique to enhance the performance of DL models on local datasets. Padarian et al. (2019a) used TL to localise a global soil vis-NIR model for prediction at local scales. TL was able to lower the error in the prediction of local data in up to 90% of the cases. In soil moisture prediction, Li et al. (2021) applied TL to improve the predictability (reduced error by up to 30%) of DL models derived from the latest SMAP dataset using the ERA5-land dataset, which has a longer time span.

Tasmania presents an ideal case study due to its diverse soils and unique climate, supporting both agriculture and biodiversity (Cotching, 2018; Cotching et al., 2009). While digital soil assessments have been conducted in Tasmania for irrigation and land management (Kidd et al., 2015b), there is a need for high resolution soil moisture maps to monitor soil water content within the profile (Kidd et al., 2015a; Kidd et al., 2014). This study aims to generate near real-time daily soil moisture maps at a 80 m, providing detailed spatial information for agricultural and environmental applications. Given Tasmania's limited point observations of soil moisture, the study explores the feasibility of applying transfer learning techniques in DL. We hypothesise that transfer learning from models trained on Australia-wide data can enhance soil moisture predictions accuracy in Tasmania. Specifically, this paper's contributions include:

(i) a systematic evaluation of DL algorithms to identify the most effective approaches for downscaling SMAP datasets to finer spatial resolution,

(ii) the innovative application of transfer learning in DL, utilising Australia-wide data to enhance soil moisture prediction accuracy in data-scarce regions like Tasmania,

(iii) comprehensive validation of the Tasmania-specific soil moisture map, providing a benchmark for future studies in areas with sparse observational data, and

(iv) the study further presents a demonstration of the feasibility of delivering live, daily soil moisture predictions, highlighting potential real-time applications in precision agriculture, water resource management, and environmental monitoring. Overall, this study addresses the current data gap by proposing scalable methods for soil moisture prediction in regions with limited observational infrastructures, thereby contributing to global efforts in sustainable land and water management.

## 2 Data and methods

### 2.1 Study area

Tasmania is an island state and Australia's southern-most territory. This area has a cool temperate climate and receives average
annual rainfall over 1500 mm in the west, and less than 600 mm in the central midlands. The rainfall variability corresponds
to its topographical features, which is characterised by rugged and high mountainous areas in the west and south-west. The
central area of the state has a large plateau with an elevation of around 1000 m above sea level (Fig 1). The midland areas are
dominated by flat lowlands (less than 290 m) for agricultural uses, with relatively small hills and mountains. Tasmania has
various soils due to the diversity of landscape, climate, and geology with Dermosols and Organosols dominating the soil types
(equivalent to Alfisols and Histosols) (Cotching et al., 2009). According to the Australian Bureau of Meteorology, soil moisture
in Tasmania was 50% in the upper soil layer (0-10 cm) and ranged from 10-85% for the root zone soil layer (0-100cm) during
the year 2022.

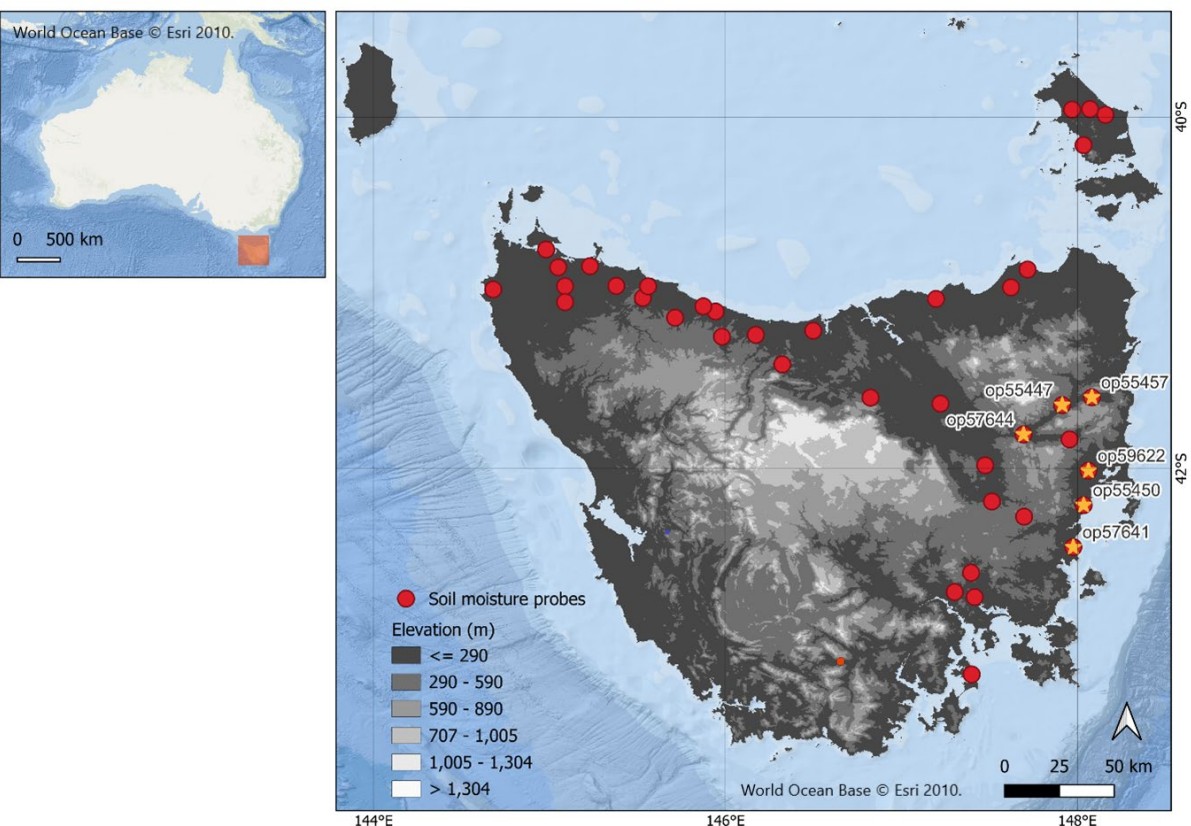

**Figure 1: Elevation map of Tasmania state. Red points represent soil moisture probes. The labelled points are stations that have**
**recorded soil moisture data for more than one year.**

## 2.2 Data sources

For the model development, we collected spatial data of parameters related to soil moisture from the Google Earth Engine database and Tasmania spatial layers. Soil moisture reference datasets were obtained from publicly available in-situ and telemetered soil moisture measurements. We separated the Australia and Tasmania datasets. The detailed information on each dataset is summarised in Table 1 and Table 2. Locations of the soil moisture stations are presented in Fig. 1 for Tasmania, and in Fig. S1 for Australia. For Australia, data were collected from the OzFlux and OzNet databases. The OzFlux provides SM data from flux monitoring tower set up to understand the exchanges of carbon and water between terrestrial ecosystems and atmosphere spreading across Australia. OzFlux stations use time-domain reflectometry Campbell Scientific sensor to record moisture level at specific soil depth that vary among the stations (see the details at https://www.ozflux.org.au/). Meanwhile, OzNet provides soil moisture records from several sites in the Murrumbidgee catchment, southern New South Wales, Australia. Each site measures soil moisture at 0-5cm with soil dielectric sensor (Stevens Hydraprobe®) or 0-8cm, 0- 30cm, 30-60cm and 60-90cm with water content reflectometers (Campbell Scientific) (Young et al., 2008).

For stations in Tasmania, soil moisture data was recorded using the capacitance (EnviroPro) probes based on frequency domain principle. Each device records moisture value at every 10 cm up to 80 cm soil depth with a frequency of 15 minutes.

**Table 1: Sources of dataset as inputs for soil moisture modelling.**

| Type of features | Group of datasets | Usage | Dataset | Spatial/ Temporal resolution | Variables (units) | Reference/source |
|---|---|---|---|---|---|---|
| Dynamic | global soil moisture data | All | SMAP L4 products (SPL4SMGP) | 9 km/ 3-hourly | sm_surface ($m^3$ $m^{-3}$) sm_rootzone ($m^3$ $m^{-3}$) | (Reichle et al., 2017) |
| | Weather data | AU | ERA5-Land | 25 km/daily | total_precipitation_sum (m) temperature_2m_min (K) temperature_2m_max (K) | (Muñoz-Sabater et al., 2021) |
| | | TAS | Weather-Now Map Tasmania | 80 m/daily | RainPrediction24hr (mm) TminPrediction (ºC) TmaxPrediction (ºC) | (Webb et al., 2020) |
| Static | Soil properties | AU | Soil and Landscape Grid of Australia (SLGA) | 90 m/- | AWC_*xxx*_EV (%) SOC_*xxx*_EV (%) CLY_*xxx*_EV (%) | (Searle et al., 2022; Wadoux et al., 2022; Malone and Searle, 2022) |
| | | TAS | Digital Soil Maps of Tasmania | 30m or 80 m/- | AWC_Tas_*xxx*_predicted_mean (%) SOC_*xxx* (%) Clay_Tas_*xxx*_mean (%) | (Kidd et al., 2015a) |

| | | | | | |
|---|---|---|---|---|---|
| Topography | All | The Shuttle Radar Topography Mission (SRTM) | 90 m/- | elevation (m) | (Jarvis et al., 2008) |
| Land use/land cover | All | Australian Collaborative Land Use and Management V8 | 50 m/- | clum_50m1218m | (Department of Agriculture Fisheries and Forestry, 2019) |
| | All | MODIS Land Cover (MCD12Q1) | 500 m/- | LC_Type1 | (Sulla-Menashe and Friedl, 2021) |

Note: *xxx* in soil datasets represent soil depth variation. Tmin = daily minimum air temperature, Tmax = daily maximum air temperature.


**Table 2: Detailed information of the soil moisture data. The location of Australian stations used in this study can be found in Supplementary Material. Number of data refers to the dataset used for models training.**

| Dataset | Source | Number of stations | Period of data coverage | Number of data |
|---|---|---|---|---|
| Australia | Oz Net | 20 | Jan 2016 – Apr 2020 | 51,411 |
| | Oz Flux | 19 | | |
| Tasmania | Ag Logic | 39 | Jan 2022 – Jul 2023 | 9,825 |

**2.3 Deep learning approaches**

In this work, we used three types of DL algorithms to develop soil moisture models: multilayer perceptron, long-short term memory, and transfer learning. These algorithms were executed in Python using *keras* in TensorFlow module (Abadi et al., 2015).

**2.3.1 Multilayer Perceptron**

Multilayer perceptron (MLP) is a type of artificial neural network consisting of hidden layers between input and output layers
(Park and Lek, 2016; Rumelhart et al., 1986). Each layer is connected by multiple perceptrons. Perceptron itself is a type of neuron that has a logical threshold in producing an output value. In MLP, the weights attached to the input of perceptrons are combined into a weighted sum and become the base value against a threshold of whether the neuron will be activated. The threshold is set by an activation function.

Since the MLP algorithm contains more than one hidden layer, combinations of perceptron between layers could resolve non-
linear relationships between input and output layers. The multilayer concept means that the perceptron's output values in one layer are propagated to the next layer as the input. At the end of the perceptron, the final output value was compared to the

reference value and evaluated using a cost function to quantify the difference between predicted and actual values. An optimization function was then used to minimise this difference metric. Additionally, this algorithm has a backpropagation scheme, which calculates the gradient error across all pairs of input and output into the first hidden layer and uses the gradient
to update the weight values. All these processes are processed in an iteration or epoch. Detailed explanations about MLP as an advanced neural network has been summarised by Huang (2009).

### 2.3.2 Long short-term memory

Long short-term memory (LSTM) is a type of recurrent neural network (RNN) that overcomes the challenge of long-term dependency in regular RNN (Zhang et al., 2021). This approach is commonly applied to sequence datasets such as time series
data (Lindemann et al., 2021). In one neuron of LSTM, there is a cell state representing the long-term memory responsible for filtering and controlling the information from input and other layers. This cell state will decide which information will be stored and passed through as output, and which information will be removed as it does not correlate to the function. There are two types of LSTM: unidirectional LSTM and bidirectional LSTM. The one-directional LSTM only stores information about the network that moves forward. Meanwhile, in bidirectional LSTM, the neural network can work in both forward and
backward directions of information flow. LSTM has been utilised for a wide range problems including soil moisture and soil temperature estimation (Li et al., 2023). The use of LSTM approach in crop yield prediction research has been thoroughly reviewed by Van Klompenburg et al. (2020) and Teixeira et al. (2023).

### 2.3.3 Transfer learning

Transfer learning (TL) is a technique in deep learning that transfers knowledge from a trained model to a new model that has
a similar architecture (Lu et al., 2015). Theoretically, the new model does not need to be trained from scratch since the transferred knowledge has an overview pattern of the data, which can reduce the training time or even increase the model's performance (Pan and Yang, 2010). A TL approach generally consists of three stages, which are (i) developing/selecting a pre-trained model, (ii) re-using the model, and (iii) fine-tuning the model. A pre-trained model can be a globally accepted general model, or a model developed based on a large dataset. Reusing the model means importing the weights of all or several layers
from the pre-trained model to the new model. Fine tuning is the training process on the transferred new model using a new specific dataset. A clear illustration of how transfer learning works is presented in Padarian et al. (2019a).

### 2.4 Soil Moisture Modelling

### 2.4.1 Data preparation

Preparing datasets for model development included data cleaning of the reference soil moisture probes data, stacking images
of covariates, and sampling the covariates based on probe locations. For the Tasmanian dataset, the recorded soil moisture data was in percentage value representing the proportion of water within the pore space in the soil. Since we need the proportion of

water within the soil volume, we converted the measured data by multiplying it with total porosity calculated from bulk density (BD) values. The BD values were derived from the digital soil map of Tasmania extracted at each probe location. Australian soil moisture dataset has been calibrated from each database source; thus, we can directly use for analysis. We then calculated the measured moisture values at various depths into an aggregated mean value for the surface (0-30cm) and subsurface (30-60cm) soil layers. We applied the spline interpolation (Bishop et al., 1999). We then calculated the average daily recorded moisture from sub-hourly records. We also converted all soil moisture data in decimals of volumetric water content ($m^3\ m^{-3}$). Covariates were collected using the Google Earth Engine platform. We first stacked weather datasets, including daily accumulated rainfall, and daily maximum and minimum temperature (TMAX and TMIN) as the reference date. Since rainfall has an extended effect on soil moisture levels, we included the current and the last 3-day rainfall data in the covariates list. Thus, we had 4 layers of rainfall data for each day ($RAIN_t$, $RAIN_{t-1}$, $RAIN_{t-2}$, and $RAIN_{t-3}$).

Daily value of SMAP soil moisture was averaged and we only selected surface (surf_SMAP) and rootzone bands (rootz_SMAP), representing 0-5cm and 0-100cm soil layers. Since SMAP-L4 products have 3-day latency, we used backward 4–7-day windows to get the sequence of SMAP bands ($SMAP_t$, $SMAP_{t-1}$, …, $SMAP_{t-n}$ with t as the day and n from 4 to 7 referring to the backward sequence). This series was then converted into a multiband image and stacked together with the weather data.

The multiband image of weather and SMAP data were then combined with land cover, elevation and spatial soil properties data. For the land cover, we used five categories, i.e. pasture, forest, rain-fed agricultural, savannah, and irrigation (PAST, FORE, AGRI, SAVA, and IRRI). FORE includes areas classified as native vegetation and native forest in CLUM or any type of forest defined in IGBP. Cropping and horticulture classes in CLUM are included in AGRI. IRRI category covers area production from irrigated agriculture and plantations in CLUM classification. SAVA includes areas defined as closed shrublands, woody savannas, and savannas in IGBP. The rest of the classes are categorised as PAST. We applied one-hot-encoding method to convert land cover categories into a binary (0s and 1s) numerical format. Each class is represented as a separate column, where a value of 1 indicates the presence of that category, and 0 indicates its absence.

For soil properties, we selected three variables that affect the water storage of soils, including available water content (AWC), soil organic carbon (SOC) and clay content (CLY). Maps representing four layers of soil depth (0-5 cm, 5-15 cm, 15-30 cm, and 30-60 cm) of each variable were incorporated as covariates. These were further named $AWC_L x$, $SOC_L x$, and $CLY_L x$, being L as layer and x as integers from 1 to 4.

Finally, the daily multiband image containing all covariates was generated based on the time frame of the Australian and Tasmanian datasets. Covariate values were then sampled at each location with measured soil moisture data, producing paired datasets of covariates and observed data for each date at every station. Any row that contained missing values in either covariates or observed data was excluded. This led to 51,411 observations covering the period of January 2016-April 2020 for the Australia dataset and 9,825 observations for Tasmania from January 2022-July 2023.

**2.4.2 Model setup**

This study set the deep learning models to have two output values representing soil moisture at the surface and subsurface layers (0-30cm and 30-60cm, respectively). The structure of the MLP model consisted of four dense layers of 128, 64, 32, and 16 neurons as the hidden layer, existing in between the input and output layers. We used Rectified Linear Unit (ReLU) and Adam optimiser as the activation and optimisation functions, respectively. The learning rate, batch size, and the number of epochs used in this algorithm were 0.0001, 128, and 150, respectively. To avoid overfitting in the training process, an early

stopping was applied based on the validation loss, which halted the training if there was no improvement after five epochs.

For the LSTM algorithm, the time series dataset of SMAP was used as input in bidirectional LSTM. This part formed a 2x8 shape, which then passed through a dense layer of 100 neurons. Combined with the rest of the covariates, this became the input of four hidden layers with 128, 64, 32, 16 neurons. To make a fair comparison, we set the activation and optimisation functions, learning rate, batch size, and the number of epochs in LSTM that are similar to the MLP.

During model training and validation, the value of $1-\rho_c$ (Lin's concordance correlation coefficient, Equation 1) was used as a loss function. We aimed its minimum value for validation to get the best model performance. The Lin's coefficient represents the distance of predicted data plotted against the observed data with the 45-degree line (Lin, 1989):

$$\rho_c = \frac{2s_{xy}}{s_x^2 + s_y^2 + (\bar{x} - \bar{y})^2} \hspace{4cm} (1)$$

where $s_x^2$ and $s_y^2$ are the variances, while $\bar{x}$ and $\bar{y}$ are the mean of the observed and the predicted SM. The $s_{xy}$ is the covariance

value was calculated using Equation 2.

$$s_{xy} = \frac{1}{n}\sum_{i=1}^{n}(x_i - \bar{x})(y_i - \bar{y}) \hspace{4cm} (2)$$

where n is the number of data, and $i$ is the order of data being calculated. This function can represent how well the model capture temporal patterns of the observed data in a time series.

For analysis, we had three scenarios for feeding these two DL algorithms. Figure 2 shows the modelling scheme used in this

study.

a.  Australia (AU) model, trained on the Australia dataset. This was based on the model developed by Fuentes et al. 2022, with a modification on feature selection as model input: (1) used the most recent product of the SMAP dataset; (2) excluded variables giving the least impact on DL predictions, which are Sentinel-1 dataset, vegetation index, and land surface temperature; (3) added daily maximum and minimum air temperature. We derived only one AU model

for each DL algorithm by splitting the dataset of 2016-2018 for training and 2019-2020 for validation.

b.  Tasmania (TAS) model, trained on the Tasmania dataset. We derived multiple models for analysis using the leave one station out cross-validation schema across 39 monitoring stations.

c.  Transfer learning (TL) model. Here, we used the trained AU model and fine-tuned the model using the Tasmanian dataset. For MLP, we transferred the weights of the first three hidden layers of the AU model and kept them unchanged

during the fine-tuning process. Meanwhile, for LSTM, we kept the first three hidden layers after LSTM output (128,

64, and 32 neuron layers) unchanged. The rest of the neurons, including the weights on LSTM architecture, were retrained.

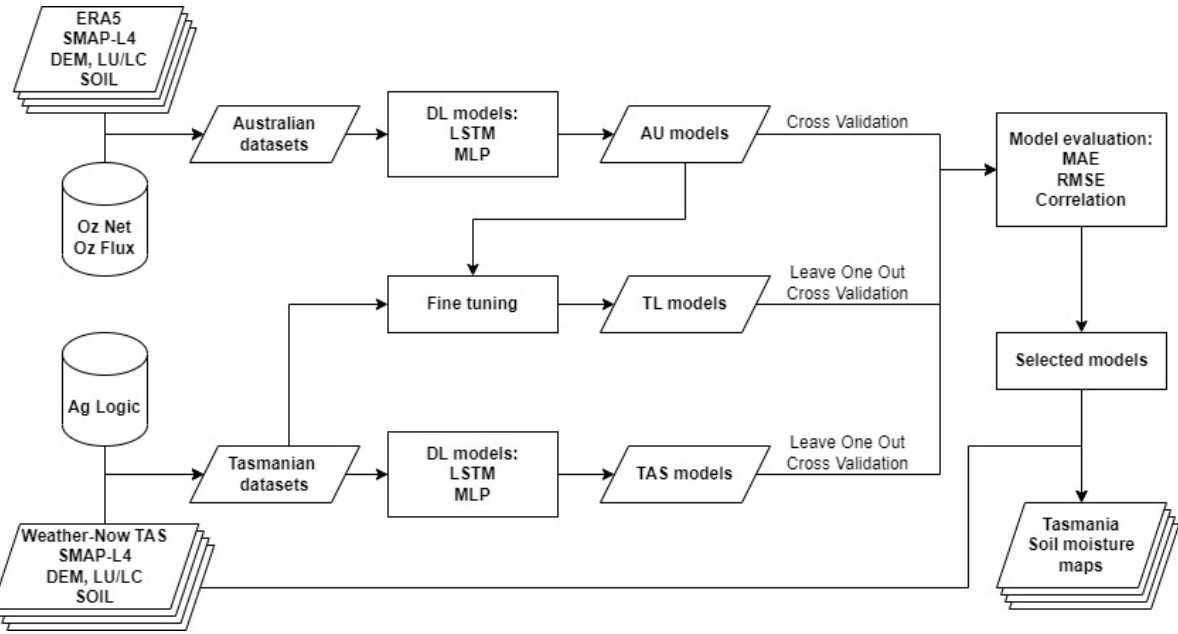

**Figure 2: Soil moisture modelling scheme.**

### 2.4.3 Model evaluation

An evaluation was first conducted on AU models. We applied AU models to predict soil moisture in Tasmania and quantified goodness of fit between predicted and measured values. Subsequently, the TAS and Transfer Learning (TL) models were evaluated using leave one station out cross validation (CV) testing scheme. This scheme comprised randomly selecting one
station as a testing set, another station as a validation set, and the rest of the stations as the training set. The scheme was applied to all probes, thus resulting in 39 models for each TAS and TL models.

The goodness of fit between prediction and observations was quantified based on mean absolute error (MAE), root mean square error (RMSE) and Pearson's linear correlation coefficient (Equations 3-5).

$$MAE = \frac{\sum_{i=1}^{n}|y_i - x_i|}{n} \tag{3}$$

$$RMSE = \sqrt{\sum_{i=1}^{n}\frac{(y_i - x_i)^2}{n}} \tag{4}$$

$$r = \frac{\sum_{i=1}^{n}(x_i - \bar{x})(y_i - \bar{y})}{\sqrt{\sum_{i=1}^{n}(x_i - \bar{x})^2}\sqrt{\sum_{i=1}^{n}(y_i - \bar{y})^2}} \tag{5}$$

where $y_i$ is moisture prediction, $x_i$ observation, and $n$ the amount of data. The final Tasmanian soil moisture maps were calculated from the average of 39 maps derived from leave-one-out-station CV schema using LSTM-TL algorithm. We also calculated the standard deviation from this model outputs to show the model's uncertainty.

### 2.4.4 Model interpretation

To explain the contribution of each input variable in soil moisture prediction, we calculated the Shapley value (Aas et al., 2021). Shapley value is the marginal contribution of each predictor after considering all possible combinations. The SHAP value is derived from the game theory and optimal Shapley values and has been widely used to interpret feature contribution in deep learning models (Padarian et al., 2020; Odebiri et al., 2022; Mohammadifar et al., 2022). In this study, SHAP calculation was based on the transferred LSTM model with a random split of 0.9:0.1 for training and testing. SHAP values resulting from the testing dataset were summed across different times or covariates for analysis. The calculation was done using the Shapley Additive exPlanations (SHAP) library in Python language (Lundberg and Lee, 2017).

## 3 Results

### 3.1 Distribution of moisture data

We first compared the soil moisture (SM) data from Australia and Tasmania datasets that were used for building the DL models. Figure 3 shows the distribution of SM data over the analysis period based on density probability and histogram plots. Tasmanian data generally had a similar pattern to that of Australian data. Both data were left-skewed for the surface layer and had a peak concentration of around $0.2 \ m^3 \ m^{-3}$. Nevertheless, Tasmanian data were slightly shifted to the right with a mean value of $0.26 \ m^3 \ m^{-3}$ higher than the Australian one (mean = $0.17 \ m^3 \ m^{-3}$). Tasmanian data ranged from 0.07 to $0.54 \ m^3 \ m^{-3}$, while Australian data ranged $0.02-0.50 \ m^3 \ m^{-3}$. Tasmanian data had a lower density value for soil moisture less than $0.2 \ m^3 \ m^{-3}$ compared to Australian data, yet they concentrated more at over $0.25 \ m^3 \ m^{-3}$.

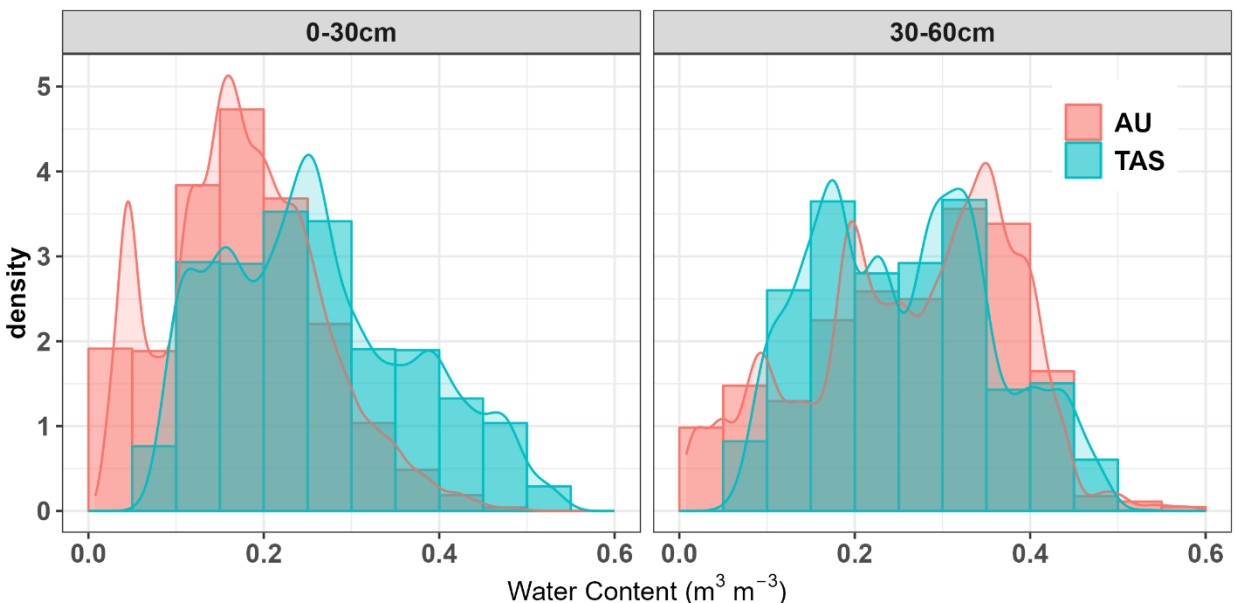

**Figure 3: Distribution plot of Australia (AU) and Tasmania (TAS) soil moisture data.**

280    Meanwhile, for subsurface soil moisture data, both regions had two peaks of data concentration (about 0.2 and 0.35 $m^3$ $m^{-3}$) yet different types of distribution. Australian data were relatively skewed to the right (skewness -0.32), while Tasmanian data were skewed to the left (skewness 0.23). The Australian data for this layer had a wider range (0.01-0.60 $m^3$ $m^{-3}$) compared to Tasmanian data (0.06-0.54 $m^3$ $m^{-3}$). Tasmanian data had more concentrations of moisture level of 0.10-0.35 $m^3$ $m^{-3}$, while the Australian data had a fair distribution of moisture levels less than 0.30 $m^3$ $m^{-3}$. Despite all the differences, both subsurface data

285    had a similar mean value about 0.26 $m^3$ $m^{-3}$.

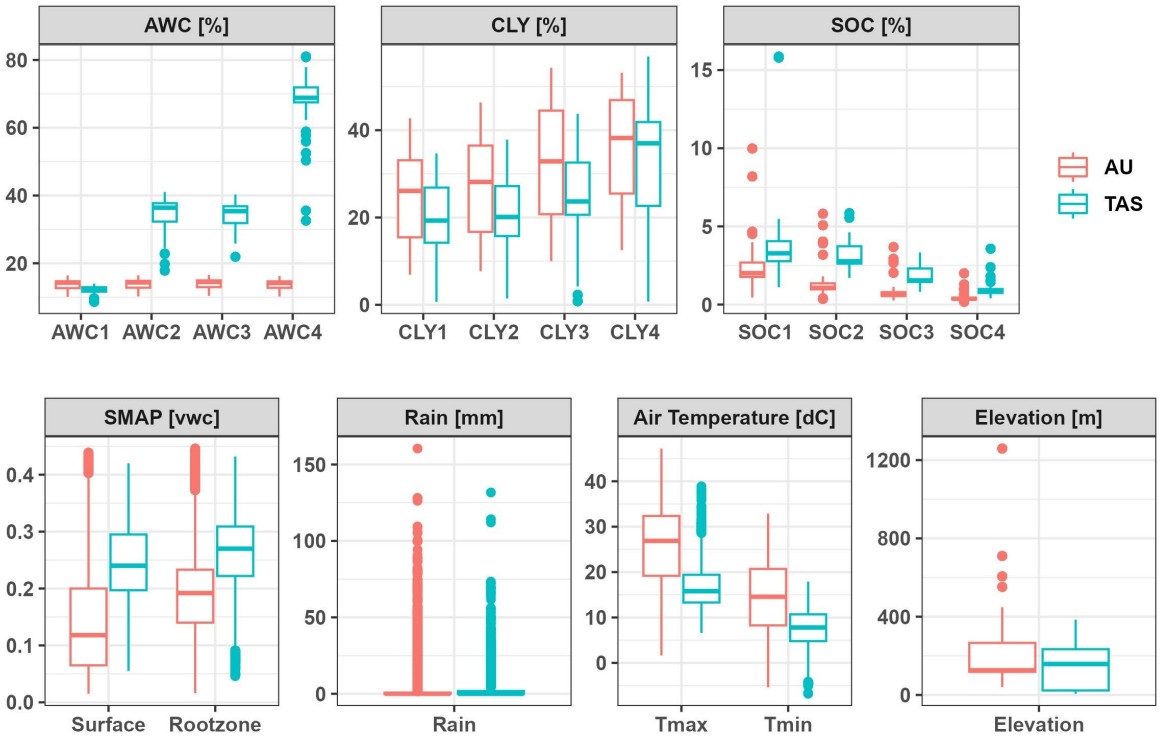

**Figure 4: Comparison of the boxplot from Australia (AU) and Tasmania (TAS) covariates used in this study, including available water content (AWC), clay content (CLY), soil organic carbon (SOC), soil moisture (in vwc [m³ m⁻³]) from SMAP, rainfall, air temperature and elevation. The number assigned next to soil properties refers to the soil layer of 0-5 cm, 5-15 cm, 15-30 cm, and 30-60 cm depths.**

We also plotted the distribution of data for each covariate extracted from Australia and Tasmania (Fig. 4). Most covariates show a distinct distribution pattern between Australia and Tasmania. Australia soil data generally had lower values on available water and carbon content, yet a higher percentage of clay content than in Tasmania. Soil moisture values extracted from the global SMAP dataset for Australia had lower mean values at both surface and subsurface soil layers, yet it had a wider range of moisture levels. For the rest of the covariates (weather data and elevation), Australian data covered a larger range of values than in Tasmania. The maximum rainfall data in Australia reached 160 mm/day, while in Tasmania, it was up to 131 mm/day. The distribution of air temperature data also followed the same trend, with Tasmania having lower mean values for both daily maximum and minimum.

### 3.2 SMAP prediction of soil moisture in Tasmania

Soil moisture content from the SMAP dataset was used as the primary covariate in our models. Thus, we first investigated the relationship between SMAP and field-observed soil moisture in Tasmania. Surface soil moisture of SMAP (0-5 cm) was

directly compared to the first level of measurement (10 cm depth), while the SMAP rootzone (0-100 cm) was against the average moisture values of all level's measurements (10-80 cm depths). The overall correlation coefficient between SMAP and measured data was 0.37 for the surface and 0.49 for the rootzone layer. SMAP SM data had a moderately high correlation coefficient with the measured data across different stations in Tasmania, with a median value 0.77 and 0.76 for surface and rootzone layer, respectively. The errors for rootzone prediction (MAE = 0.08 $m^3$ $m^{-3}$ and RMSE = 0.10 $m^3$ $m^{-3}$) were slightly lower than surface prediction (MAE = 0.09 $m^3$ $m^{-3}$ and RMSE = 0.11 $m^3$ $m^{-3}$). According to the distribution of errors and correlation coefficients across the measuring stations, SMAP of the rootzone layer had a wider range value of errors and correlation coefficients compared to the surface layer (Fig. 5). In addition, there were more stations with negative correlation values for the rootzone SMAP.

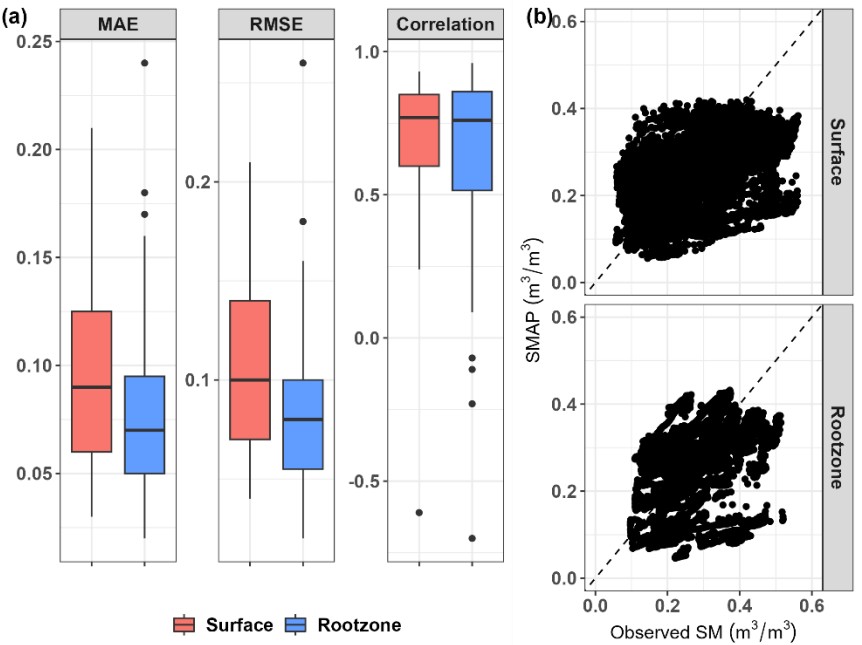

**Figure 5: Performance of soil moisture derived from SMAP dataset compared to measured data in Tasmania during the period of January 2022-April 2023: (a) the distribution of mean absolute error (MAE), root mean square error (RMSE), and correlation valued at each probe location, (b) overall performance in scatter plot between predicted and measured soil moisture data compared to the 1:1 line (dashed line).**

### 3.3 Model selection and performances

We tested the ability of Australian models to predict SM in Tasmania. In general, models with the MLP approach performed better than LSTM for both surface and subsurface layers, with MLP average MAE = 0.1 $m^3$ $m^{-3}$, RMSE = 0.12 $m^3$ $m^{-3}$ and correlation = 0.49 compared to LSTM average MAE = 0.12 $m^3$ $m^{-3}$, RMSE = 0.15 $m^3$ $m^{-3}$ and correlation = 0.48 (Fig 6). The MLP model resulted in predictions that were closer to the 45-degree line with the observed data. Furthermore, according to the distribution of performance across Tasmanian stations, the MLP model predictions had lower errors and less variation, as

shown by the boxplot. The LSTM model had good correlations (>0.6) in most stations. However, despite the good results of the MLP algorithm, there was no improvement in the prediction accuracy when compared to using just the SMAP dataset (Fig 5).

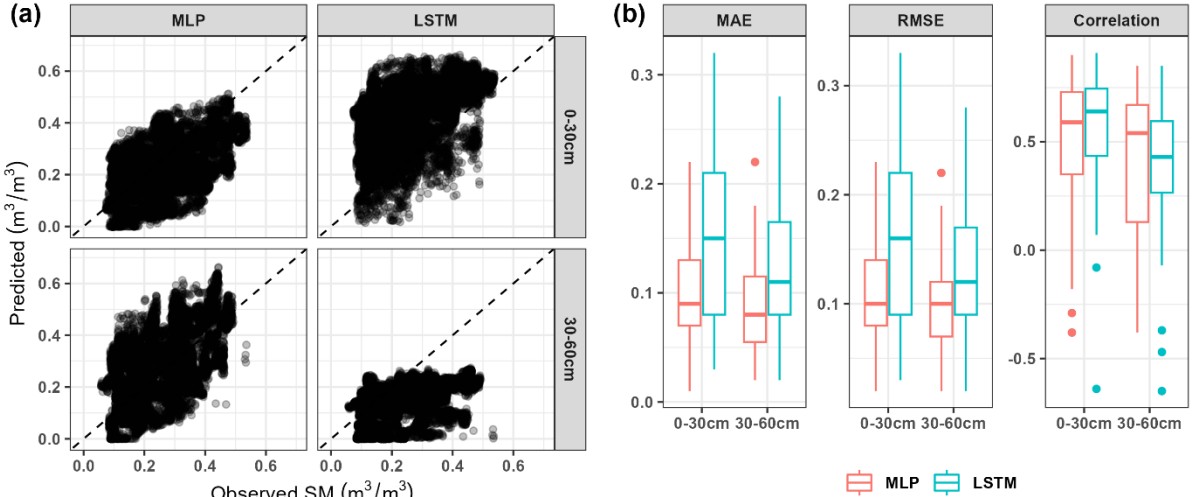

**Figure 6: Performance of Australian models for soil moisture prediction in Tasmania based on multilayer perceptron (MLP) and long-short term memory (LSTM) approaches: (a) overall comparison between predicted and observed soil moisture data, (b) distribution of mean absolute error (MAE), root mean square error (RMSE) and correlation value across 39 stations in Tasmania.**

Thus, the second set of models was trained on Tasmanian data using the leave one station out cross-validation scheme. The results show that the predicted soil moisture varied from 0 to 0.8 $m^3$ $m^{-3}$, giving a larger range value than the observed data (Fig 7). The scatter plots of predictions and observations show a large dispersion, with some zero value predictions regardless of the variation of the observed data. Both DL approaches had similar results in performance valuation. The MLP models were slightly better than LSTM, with average MAE = 0.12 $m^3$ $m^{-3}$, RMSE = 0.15 $m^3$ $m^{-3}$ and correlation = 0.43 for MLP models, while the LSTM models had MAE = 0.13 $m^3$ $m^{-3}$, RMSE= 0.17 $m^3$ $m^{-3}$, and correlation = 0.26. Model evaluation on each station showed that error values and correlation of both DL models for subsurface soil moisture prediction (0.01-0.48 $m^3$ $m^{-3}$ for MAE and RMSE; -0.63 to 0.96 for correlation) were more varied compared to surface moisture predictions (0.02-0.35 $m^3$ $m^{-3}$ for MAE and RMSE; -0.07 to 0.94 for correlation).

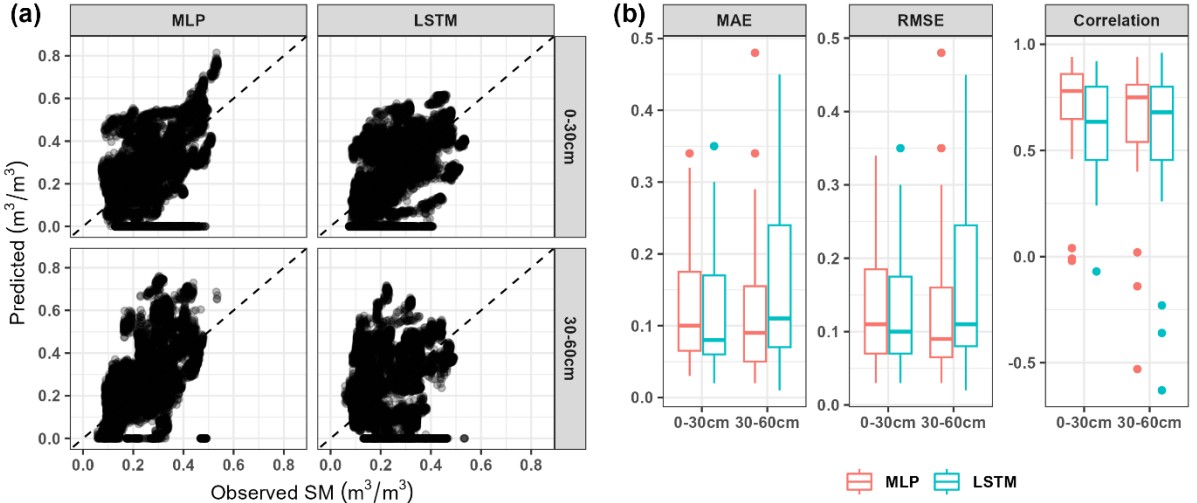

**Figure 7: Performance of Tasmanian models for soil moisture prediction in Tasmania based on multilayer perceptron (MLP) and long-short term memory (LSTM) approaches: (a) overall comparison between predicted and observed soil moisture data, (b) distribution of mean absolute error (MAE), root mean square error (RMSE) and correlation value across 39 stations in Tasmania based on leave-one-out cross validation scheme.**

Finally, the transfer learning approach was deployed by transferring knowledge from the trained Australian models into Tasmania. Visually, data points resulting from TL models against the observed data were closer to the 45-degree line for both MLP and LSTM (Fig 8). The predicted data of MLP were in the range 0 up to 0.7 $m^3$ $m^{-3}$, being larger than that of LSTM (0.03-0.63 $m^3$ $m^{-3}$). The overall performance of LSTM models was MAE = 0.07 $m^3$ $m^{-3}$, RMSE = 0.08 $m^3$ $m^{-3}$, and correlation = 0.73. This was slightly better than the performance of the MLP models, with average MAE, RMSE and correlation of 0.08 $m^3$ $m^{-3}$, 0.09 $m^3$ $m^{-3}$, and 0.62. The distribution of model performance for both DL algorithms on predicting soil moisture across all stations in Tasmania was quite similar. However, the LSTM model with transfer learning had a more consistent performance for the surface and subsurface layer, as shown by the upper quartile of the boxplot for errors. This infers that most stations had error values less than 0.08 $m^3$ $m^{-3}$ for surface and subsurface predictions.

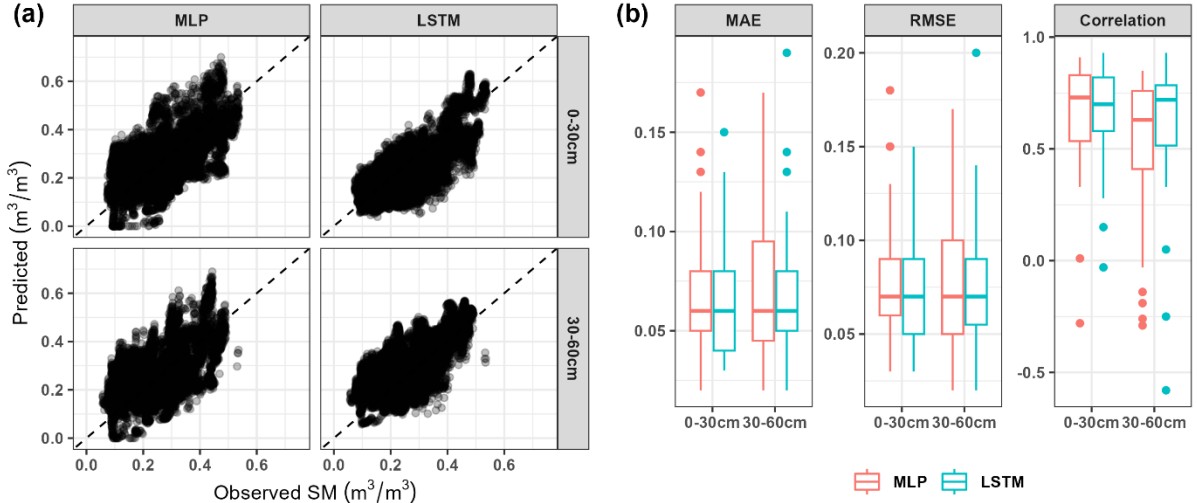

**Figure 8: Performance of transfer learning models for soil moisture prediction in Tasmania based on multilayer perceptron (MLP) and long-short term memory (LSTM) approaches: (a) overall comparison between predicted and observed soil moisture data, (b) distribution of mean absolute error (MAE), root mean square error (RMSE) and correlation value across 39 stations in Tasmania based on leave-one-out cross validation scheme.**

Comparing the performance of the six models for predicting SM in Tasmania, it becomes evident that the LSTM with transfer learning approach (LSTM-TL) was optimal. We further analysed its performance according to station locations, time series, land cover types, and seasonal time.

The spatial distribution of the performance of LSTM-TL model using different stations, is shown in Fig. 9. Stations with high correlation values (> 0.74) mostly corresponded to lower error, with RMSE values less than 0.087 $m^3 m^{-3}$. Meanwhile, stations with large errors (RMSE > 0.106 $m^3 m^{-3}$) had moderate to high correlation coefficients (>0.55). In addition, the stations with the lowest correlations had RMSE ranging from 0.068 to 0.106 $m^3 m^{-3}$.

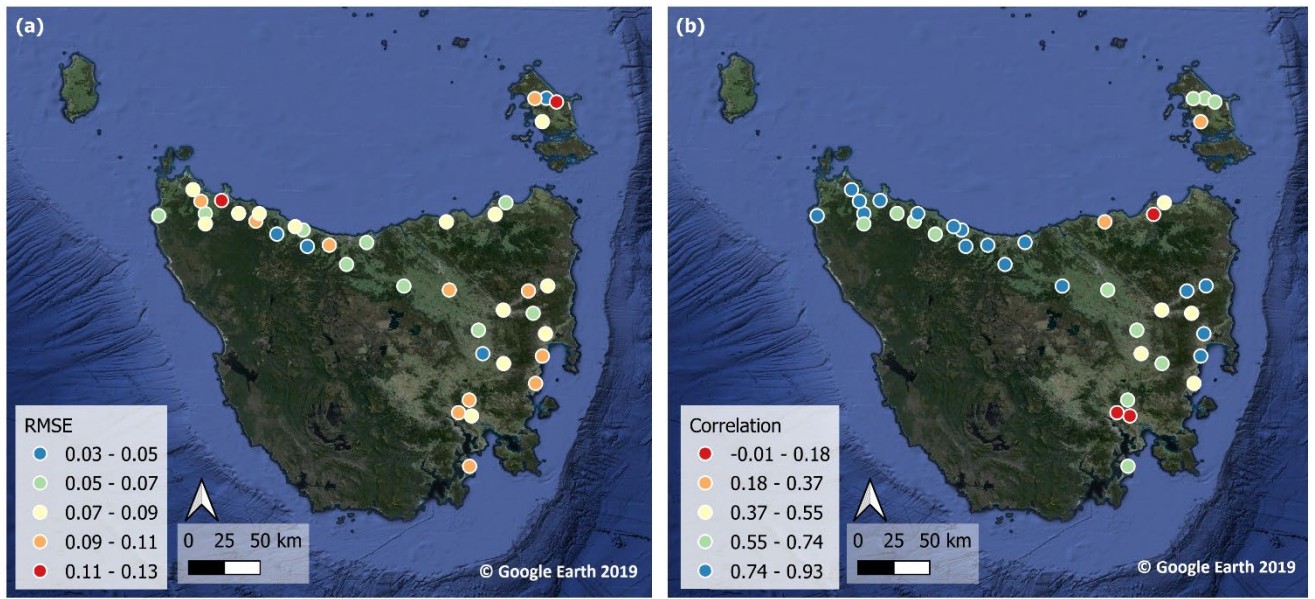

**Figure 9: Spatial distribution of the performance of Long short-term memory (LSTM) model with transfer learning for predicting soil moisture at each station in Tasmania. The evaluations are in average of: (a) root mean-square error (RMSE) and (b) Pearson's correlation coefficient across surface (0-30cm) and subsurface (30-60cm) soil layers.**

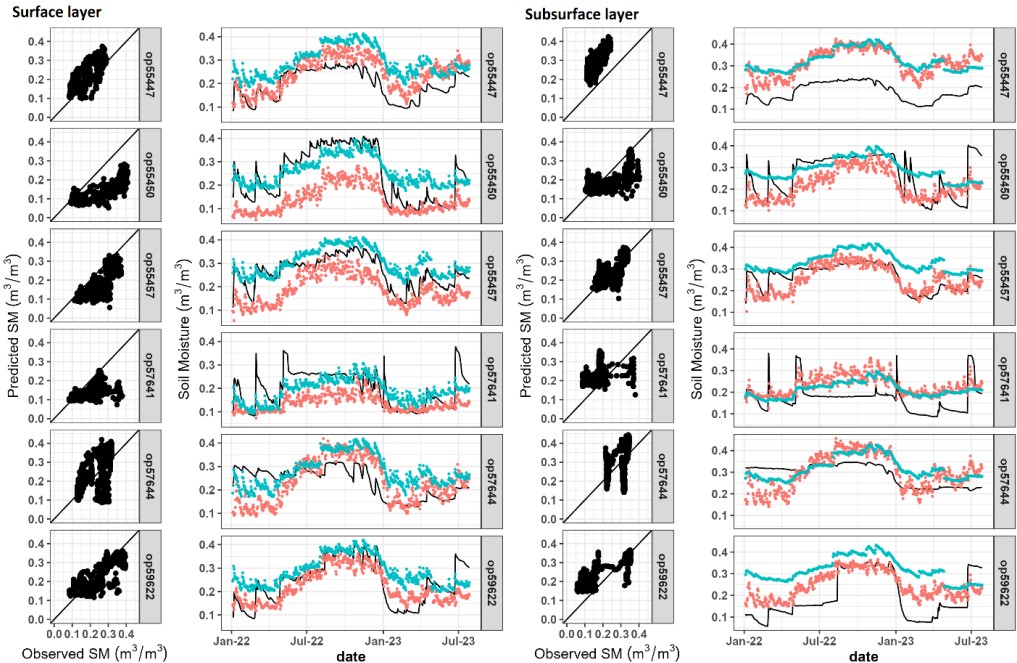

**Figure 10: Performance of models resulted from leave one station out validation scheme for six stations with the longest observation period: op55447, op55450, op55457, op57641, op57644, and op59622. The right panel shows the prediction of the entire series (red dots) compared to SMAP predictions (blue dots) and the observed data (black line). Note that SMAP predictions in the surface panel represent 0-5 cm, while the subsurface panel refers to 0-100 cm.**

Time series predictions for six typical stations compared to SMAP and observed data are plotted in Fig. 10. These cases show that our model predictions follow the dynamics of the observed data, with correlation coefficients varying from 0.43-0.84 for the surface layer, and 0.35-0.85 for the subsurface layer. Our moisture predictions were relatively lower than the value from

SMAP, yet the predictions better matched the observed data.

Table 3 highlights our model performance based on seasonal variations. The most accurate performance was achieved during summer, with an average correlation coefficient up to 0.72 and RMSE values around 0.06 $m^3$ $m^{-3}$. In other seasons, our model performed at MAE values ranging from 0.045 to 0.079 $m^3$ $m^{-3}$, with RMSE at 0.052 to 0.082 $m^3$ $m^{-3}$. Spring was identified as having a low correlation at both soil layers.


Table 3: Model performance during four seasons in Tasmania. The values were aggregated from all stations. MAE = mean absolute error, RMSE = root mean square error.

| Season | Soil layer | MAE ($m^3$ $m^{-3}$) | | RMSE ($m^3$ $m^{-3}$) | | Correlation coefficient | |
|---|---|---|---|---|---|---|---|
| | | mean | std | mean | std | mean | std |
| Autumn | 0-30cm | 0.060 | 0.033 | 0.071 | 0.034 | 0.302 | 0.296 |
| | 30-60cm | 0.077 | 0.039 | 0.084 | 0.039 | 0.242 | 0.351 |
| Spring | 0-30cm | 0.079 | 0.036 | 0.082 | 0.035 | 0.095 | 0.227 |
| | 30-60cm | 0.045 | 0.040 | 0.052 | 0.038 | 0.098 | 0.300 |
| Summer | 0-30cm | 0.058 | 0.021 | 0.065 | 0.023 | 0.674 | 0.322 |
| | 30-60cm | 0.066 | 0.032 | 0.075 | 0.031 | 0.723 | 0.211 |
| Winter | 0-30cm | 0.067 | 0.037 | 0.072 | 0.037 | 0.368 | 0.294 |
| | 30-60cm | 0.070 | 0.044 | 0.075 | 0.043 | 0.275 | 0.393 |

We also checked how our selected model performs in different land use categories (Table 4). Overall, the prediction

consistently resulted in error values of 0.06 up to 0.09 $m^3$ $m^{-3}$ and correlation coefficients between 0.51 and 0.76 for both soil layers. Soil moisture prediction on the pasture area performed best with the least error values (RMSE = 0.07 $m^3$ $m^{-3}$), with a high correlation coefficient (0.62). While forested area (with less number of stations) had the lowest correlation (0.550 and 0.623 for surface and subsurface) followed by savannah (0.598 and 0.511 for surface and subsurface).


**Table 4: Performance of the selected model for predicting soil moisture at both soil layers aggregated by land use/land cover class (mean and standard deviation). MAE = mean absolute error, RMSE = root mean square error, n = the number of stations.**

| Land use category | Soil layer | MAE ($m^3$ $m^{-3}$) | | RMSE ($m^3$ $m^{-3}$) | | Correlation | | n |
|---|---|---|---|---|---|---|---|---|
| | | mean | std | mean | std | mean | std | |
| Forest | 0-30 cm | 0.087 | 0.035 | 0.093 | 0.031 | 0.550 | 0.234 | 3 |
| | 30-60 cm | 0.080 | 0.010 | 0.083 | 0.012 | 0.623 | 0.176 | 3 |
| Irrigation | 0-30 cm | 0.063 | 0.039 | 0.072 | 0.038 | 0.769 | 0.151 | 11 |
| | 30-60 cm | 0.077 | 0.049 | 0.085 | 0.050 | 0.736 | 0.149 | 11 |
| Pasture | 0-30 cm | 0.059 | 0.023 | 0.069 | 0.027 | 0.657 | 0.244 | 13 |
| | 30-60 cm | 0.060 | 0.030 | 0.072 | 0.032 | 0.576 | 0.317 | 13 |
| Savannah | 0-30 cm | 0.065 | 0.033 | 0.074 | 0.029 | 0.598 | 0.206 | 12 |
| | 30-60 cm | 0.071 | 0.023 | 0.079 | 0.025 | 0.511 | 0.399 | 12 |

### 3.4 Spatial pattern of predicted soil moisture

We then applied our calibrated models to predict soil moisture for the whole area of Tasmania at a daily time step, and then aggregated the values for each season (Fig. 11). High soil moisture occurred in the western part of Tasmania, and small forested areas in the northeast. However, the western part was predicted as the driest area at the subsurface layer in all seasons. Our models estimated subsurface soil moisture at 0.01 to 0.55 during the summer-autumn, and up to 0.62 during the winter-spring. The average of standard deviation maps varied up to 0.08 for both soil layer predictions. In most of the high moisture level areas (near 1 $m^3$ $m^{-3}$), the deviation maps show the lowest value for surface moisture prediction. Higher deviation value was identified in the central highland areas and hilly regions in the east and northeast. Meanwhile, the deviation map for subsurface soil moisture prediction depicts a higher uncertainty model over the western part of the state.

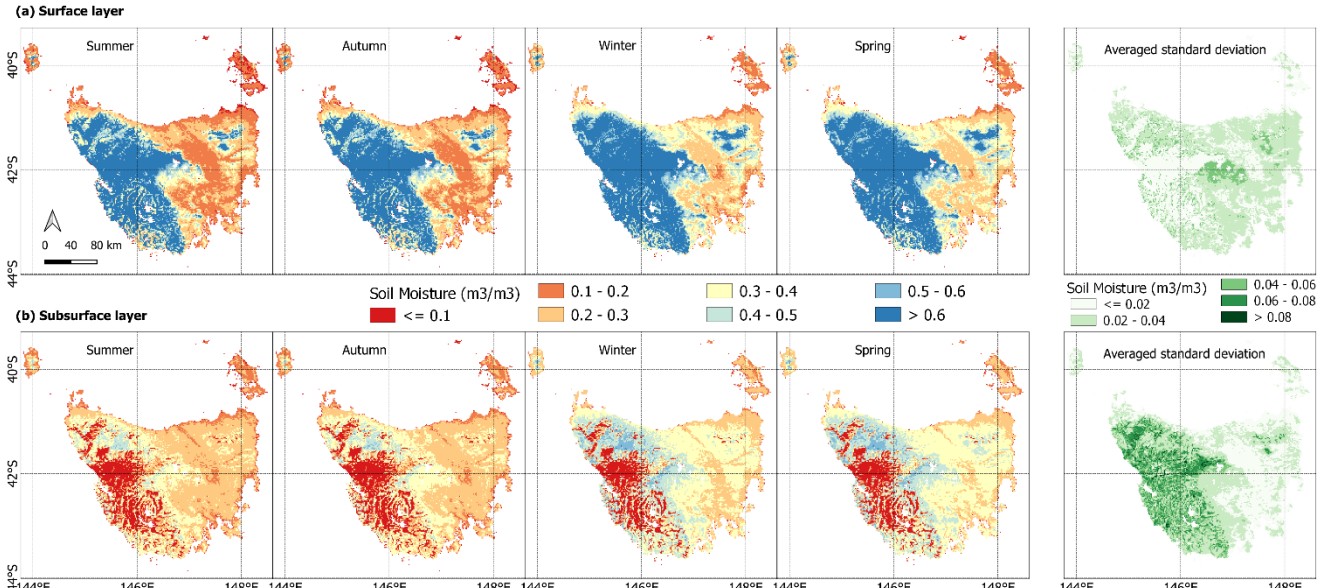

**Figure 11: Spatial pattern of seasonal average predicted soil moisture along with its averaged standard deviation in Tasmania for (a) surface (0-30 cm) and (b) subsurface (30-60 cm) layers using LSTM models with the transfer learning approach. Soil moisture values are in m³ m⁻³.**

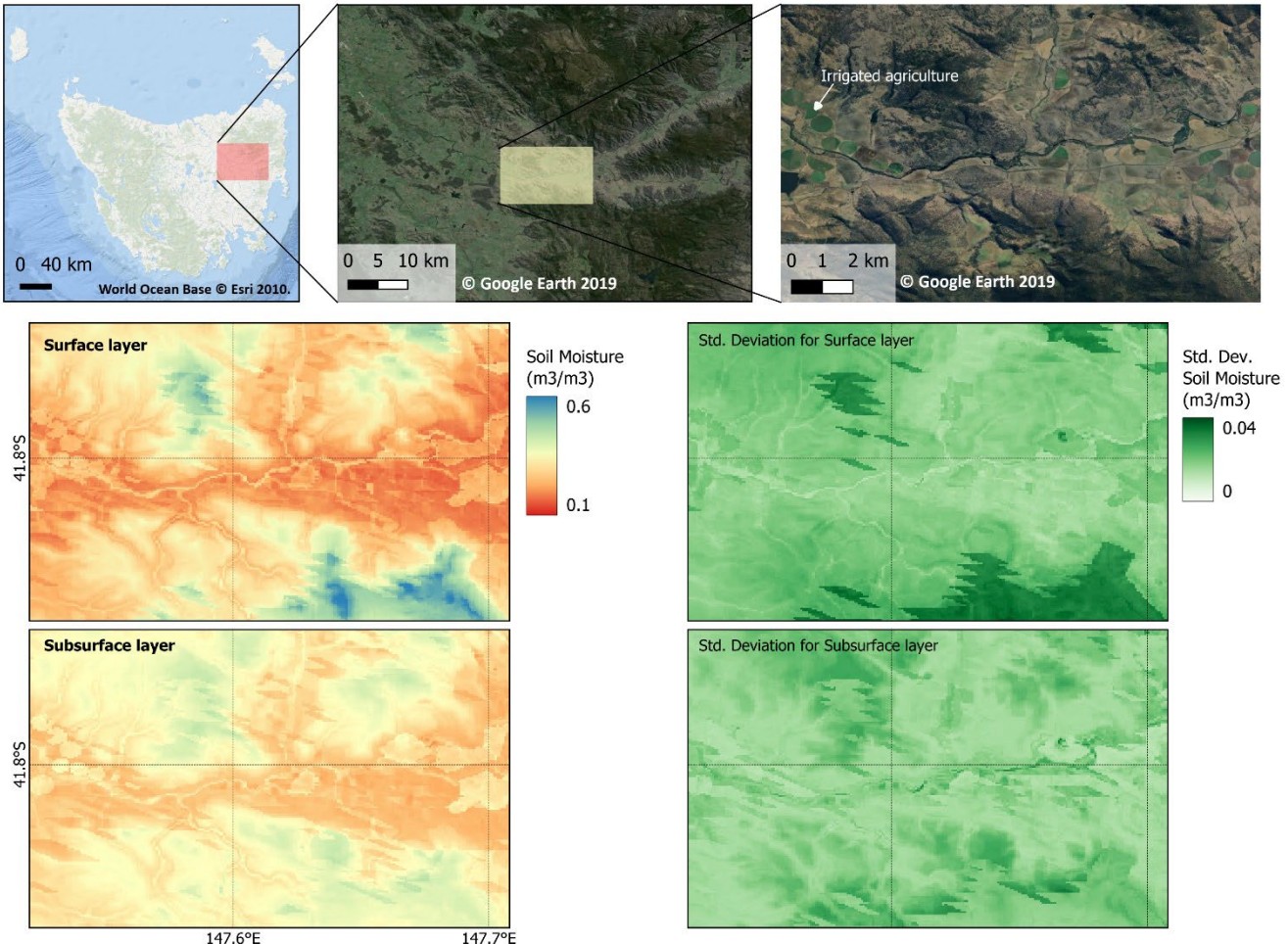

**Figure 12: Soil moisture predictions and its standard deviation for surface and subsurface layers on the date of 2023-09-10 as an example of 80 m resolution map. The zoomed panel represents an area of the Fingal Valley.**

An example of the 80 m resolution soil moisture maps for each soil layer and their uncertainty values over an area in the eastern part of Tasmania is given in Fig. 12. The Fingal Valley region encompasses agricultural lands with irrigation systems, including identifiable center pivot systems, distributed along the river between mountainous areas. The surface soil moisture map effectively captured topographical variations, as indicated by distinct colour differences between the mountainous areas and their surroundings. Agricultural areas had lower moisture values (orange colour), whereas higher values were predicted in mountainous areas. The uncertainty values were mostly less than 0.025, except for the high elevation area. Similarly, subsurface predictions can represent the spatial variation of the area of interest, particularly in irrigation areas and rivers. The uncertainty was more varied than the surface prediction, with no clear spatial pattern.

### 3.5 Features contribution

The importance of each input variable on LSTM transfer learning model outputs was analysed using the SHAP value. The violin plot (Fig. 13) summarises three pieces of information: (1) overall comparisons in feature importance, (2) distribution and variability SHAP value of each feature (3) the value of the feature shown by colour scaling from Low to High. Based on the testing dataset (n = 884), it indicates that SMAP dataset was the most important feature in predicting both surface and subsurface soil moisture. These were followed by LULC and soil properties (SOC and clay content). Elevation and weather

data, including temperature and rainfall were the least important covariates in our models. SMAP surface had the widest range of SHAP values varying from -0.25 to 0.35. A high density of SMAP SM surface occurred in negative SHAP values, implying a reducer of the model output. High soil moisture in SMAP surface gave additional value to the output prediction. However, SMAP rootzone had a reverse pattern, with a fair distribution of SHAP value ranging from -0.2 to 0.2, high SM in SMAP rootzone negatively impacted the model output, and vice versa. Other covariates had less impact on the model output with

SHAP value within -0.1 to 0.1. Land use and daily minimum air temperature predominantly gave positive impact on the output.

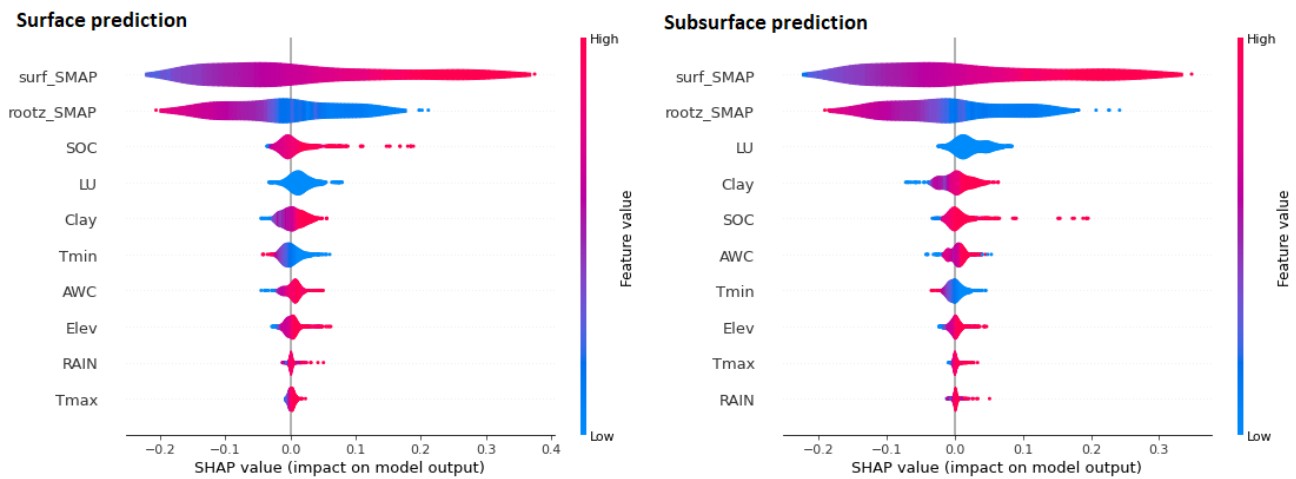

**Figure 13: Aggregated SHAP value for each input dataset representing its impact on surface (left) and subsurface (right) soil moisture prediction based on LSTM with transfer learning model.**

## 4 Discussion

### 4.1 MLP and LSTM approaches

We compared the MLP and LSTM as modelling algorithms to predict surface and subsurface soil moisture simultaneously. Our results revealed that MLP outperformed LSTM when we directly applied to the Australian models to predict Tasmania soil moisture, yet contradictory results were found when using TL models. Nevertheless, both algorithms with TL approach

were equally good in predicting SM (Fig. 8). In the case of Australian models, the LSTM only record the 'memory' of how

the previous moisture and rainfall changes daily soil moisture in Australia. When the LSTM was directly used to process SMAP in Tasmania, the 'memory' of Australian data might not apply in Tasmania, causing a larger error. In transfer learning, we let the weight of each cell in LSTM change during a fine-tuning process. This means that the model can update its 'memory' of daily SMAP according to the Tasmanian dataset.

We chose the LSTM approach as our final model as it provides consistent results in predicting surface and subsurface soil moisture. Fuentes et al. (2021) compared the performance of LSTM and MLP in Australia. Their MLP models resulted in a slightly lower error compared to the LSTM, yet they chose the concatenated LSTM over standalone MLP as the recurrent neural networks could capture the delayed effect of soil moisture change occurring between soil layers. Another research comparing LSTM and MLP to forecast soil moisture up to 6-day ahead at multilayers of soil showed that LSTM model

consistently resulted in a lower RMSE value (less than 0.09) (Han et al., 2021). However, we noted that their study used one output value for each soil layer, not implementing simultaneous predictions. Additionally, the LSTM approach has been widely investigated to model soil moisture with reliable performances in terms of spatial, time-series, and forecast analysis (Li et al., 2022a; Park et al., 2023; Fang and Shen, 2020; Datta and Faroughi, 2023).

## 4.2 Comparing Australia, Tasmania and transfer learning models

Based on our three scenarios, Australia models (AU) performed worst regardless of the type of deep learning approach. High error in AU predictions was likely due to the different distribution of datasets between Australia and Tasmania. The results also showed that the direct application of deep learning models in other local areas requires data similarity consideration. Comparing the performance of the Tasmania (TAS) and the transfer learning (TL) models, we found that TL models resolved the drawback of the TAS model, which could not fully capture the variations of the Tasmania dataset. As illustrated in Fig. 7,

the TAS models exhibited shortcomings in predicting soil moisture, notably yielding zero values in some conditions. This outcome suggests that based on data from 37 stations, the model's training was inadequate in encompassing the full range of variability within the testing dataset. Consequently, this limitation hindered the TAS model's capacity to estimate soil moisture values when confronted with input values that extend beyond the scope of the training dataset. The small sample size in the training dataset may have limited the model's ability to generalise over Tasmania's major landscapes, topographical features,

and soil properties.

To address this issue, the Transfer Learning (TL) models effectively assimilated knowledge from the more extensive Australian dataset, resulting in a substantial enhancement in the performance of the TAS model. This approach significantly enhanced the training of the TL model, as it only required adjustments to the previously learnt weight values to align them with the characteristics of the Tasmania dataset. In contrast, the TAS model required a complete training process from scratch, with

random values assigned to the weights of the deep learning (DL) layers as the initial conditions.

Adopting a transfer learning approach has shown significant potential for enhancing both training effectiveness and model performance. Our Transfer Learning (TL) models, in particular, exhibited remarkable performance improvements, surpassing the TAS models by a factor of two. This translated to error reductions of up to 45% and a 50% increase in correlation

coefficients. Furthermore, these enhancements were consistently reflected in the accurate prediction of both surface and subsurface soil moisture levels.

The efficacy of transfer learning has been explored for several applications, for example (Li et al., 2021) demonstrate that employing transferred Deep Learning (DL) models based on ERA5-land data led to a substantial increase in the explained variation of observed data, exceeding 20% in some areas of China. (Padarian et al., 2019a) also reported that the transferred local model, designed for predicting soil properties from infrared spectra, outperformed both individually trained global and local models.

## 4.3 Spatiotemporal variation of predicted soil moisture

Soil moisture maps for Tasmania were generated using the LSTM with transfer learning models (Fig. 12). At an 80-meter resolution, the model's performance is on par with the original models designed for 90-meter soil moisture predictions in Australia (Fuentes et al., 2022). A good performance of the models on irrigation land use area (Table 4) showed the ability of DL algorithm to adjust prediction value based solely on the logical value of where the station is located. This could be an alternative for modelling over irrigated areas with a lack of irrigation records. Nevertheless, there were still some limitations. While the map effectively captured the SM variation of the eastern part of Tasmania, our predictions still struggled to capture the variability of SM in the rocky, mountainous areas in the western part of Tasmania. This limitation is due to the absence of observational data in these remote regions, meaning that our model lacked the necessary information to learn and make accurate predictions. Additionally, some stations with less than one year of observational data could give strange results when evaluating models' performance. This is shown by some stations, mainly with observational data less than three months, that also have high correlation yet high error (Figure 9). This could also confirm that future research should update this modelling effort as more SM data being recorded.

Furthermore, upon comparing the soil moisture maps with the input raster dataset used for model training, the western part of Tasmania has soil organic carbon (SOC) content exceeding 20% (Kidd et al., 2015a). Additionally, the region's high altitude, exceeding 890 meters, was discernible from the elevation image (see Fig. 1). These peatlands with high SOC levels surpassed the maximum value of SOC present in our training dataset, which had a maximum of 15%. As a consequence, our models produced very high SM values (near 1) for surface predictions and small values (near 0) for subsurface predictions. Additionally, the SHAP value indicated that soil organic carbon (SOC) contributed significantly to the SM prediction in this area, overshadowing the contribution of the SMAP dataset (refer to Fig 14). Those results align with the low predicted soil thickness (< 50 cm) across western Tasmania (Kidd et al., 2015a), which certainly contributed to the low moisture level at the subsurface layer.

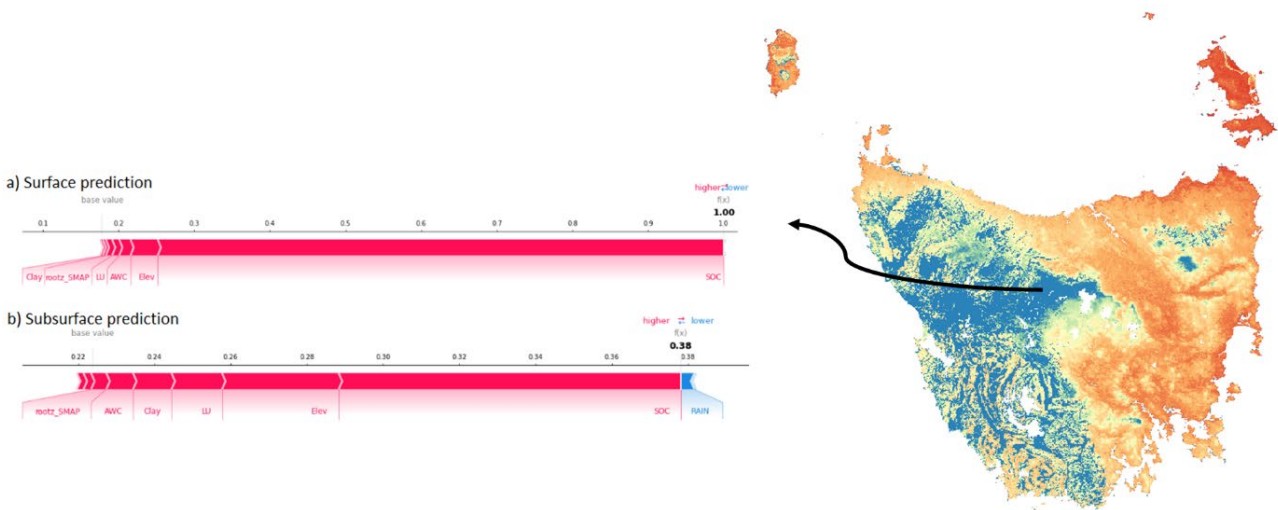

**Figure 14: An illustration of feature contribution in generating soil moisture prediction at remote area. Base value represents the average of model output over the training set specifically for SHAP analysis, while f(x) is the final prediction of soil moisture value.**

### 4.4 Assumptions and limitations

While we demonstrated the ability of the transfer learning model to accurately predict soil moisture using leave one station out testing protocol, we recognise some assumptions and limitations of the study. We assumed that our reference data represent
real moisture level values in each soil layer, however there are possible biases from the interpolation and calibration procedure on recorded data from the probes. Moreover, the limited stations (6 out of 39) that cover soil moisture dynamics of more than one year of records may not sufficiently capture the overall temporal and spatial variation of moisture in Tasmania. In addition, we believe that our cross-validation scheme has not sufficiently covered all the spatial and temporal dimensions of soil moisture prediction.

### 4.5 Future work

In this research, we only tested two algorithms, namely LSTM and MLP, which are combined with transfer learning techniques. Other DL algorithms could improve soil moisture maps' accuracy at fine resolutions in Tasmania. For example, the input covariates could include spatial context represented as images using convolutional neural networks (Padarian et al., 2019b). Our models could further consider several remote sensing data which are commonly used as covariates in soil moisture
mapping, such as vegetation index and surface temperature (Xu et al., 2022; Zhao et al., 2022; Xu et al., 2021). Furthermore, feature selection as the input for models can be explored further to derive better model performance.

However, a major consideration in this study lies in the need to incorporate a greater number of field-measured stations covering unrepresented regions, thereby enhancing the spatiotemporal representation of the data. As additional data becomes available from the existing soil moisture stations, there exists the opportunity to refine the model even further, enabling it to

capture a more comprehensive range of temporal variations. In addition, incorporation of process-based models would enhance the prediction and also allow for soil moisture forecasting (Liu et al., 2022; Minasny et al., 2024).

## 5 Conclusions

This study addresses the issue of using DL for mapping soil moisture in Tasmania given limited training datasets. Transfer learning within the deep learning framework has become a prevalent technique for enhancing model performance. This
approach was successfully applied to estimate daily soil moisture levels in Tasmania. In this context, a pre-trained soil moisture model, initially derived from the Australian dataset, serves as the reference.

The transferred models tailored for Tasmania had a superior performance in predicting soil moisture from the surface to a depth of 60 cm, all at an 80-meter resolution. When combined with the LSTM algorithm, transfer learning effectively doubles the performance compared to non-transferred models. These enhancements signify that the transferred LSTM models can be
effectively employed for daily monitoring of soil moisture levels throughout Tasmania.

The model is now available live at: https://sdi.tas-hires-weather.cloud.edu.au/shiny/ predicting soil moisture at a daily interval along with weather information (rainfall, temperature), potentially enabling land managers and farmers to make informed decisions on managing soil water for crop production and environmental monitoring.

## 6 Code Availability

Code for integrating the optimal model into near real time monitoring is available at GitHub repository (https://github.com/marliana-widyastuti/sm-map-tas.git). Data will be made available upon request.

## 7 Competing Interests

The contact author has declared that none of the authors has any competing interests.

## 8 Acknowledgements

This research was supported by ARC Discovery project Forecasting Soil Conditions DP200102542. The computation used the Nectar Research Cloud, a collaborative Australian research platform supported by the NCRIS-funded Australian Research Data Commons (ARDC). MW was funded by Lembaga Pengelola Dana Pendidikan (LPDP) Scholarship (LOG-7157/LPDP/LPDP.3/2023). We thank Ag Logic and NRM South who have allowed us access to their soil probe network to conduct this research.

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
