# Peer review of "Mapping near real-time soil moisture dynamics over Tasmania with transfer learning"

_EGUsphere, 2024_

## Author Comment (AC1)

**Reviewer 1**

This was an interesting bit of research for me to review. I understood the nature of the modelling in terms of having large national model and then adjusting it to make it more useful at local level (here across Tasmania) through using transfer learning. For research's sake the authors also tested extension of Australian model to Tasmania, which as a null hypothesis would be clearly rejected. Local models better, but transfer learning, better still. There is merit in this work but my misunderstandings of it come from the data inputs, mainly the observational data. I think these data were not well described and there is little to go on about how they were sourced. Some of the inputs into the model could be better too and which are also publicly available. I made many comments and feel there is more work to do in places. Am not a fan of the introduction for example which just seems to be a disparate collection of things in much need of pulling together into a single narrative.

Thank you very much for your positive feedback to our work.

Abstract. Last sentence needs re-wording

We have revised this sentence to be clearer.

Line 40-60. Might worth recognizing that in Australia there are a number of spatio-temporal models of soil moisture that generate maps continentally. Take for example, The Bureau of Meteorology's Australian Water Resources Assessment Landscape model (AWRA-L) version 6 (Frost et al. 2016). Another one is The National Soil Moisture Information Processing System (SMIPS; Stenson et al. 2021). Similarly model presented in Wimalathunge & Bishop (2019) is set up to run daily and uses inputs derived from Soil and Landscape Grid of Australia. It might also be helpful to mention a few of the popular water balance models that are out there too or incorporated into broad land surface models.

Thank you very much for your insights. We have added those references about moisture mapping in Australia: **AWRA-L maps** daily moisture across the continent at 5km resolution. this one available in BOM gov data. Available on daily basis for end user. **SMIPS dataset**: resolution at 1km showing the moisture level for two layers, upper (10 cm depth) and deeper (10 to 80 cm). Publicly available for end user, with latency of 3 days. Accuracy was increased by adjusting SMOS data in the model prediction. **Wimalathunge & Bishop (2019)** has shown the potential of creating high-res SM mapping for Australia by combining machine learning and water balance model to accurately predict daily soil moisture. This research is the backbone of the SMIPS dataset.

Line 70. Not sure i follow about "..to a model with similar tasks"

This sentence has been revised accordingly.

Line 78-81. Would be good to reference some reports here. Probably guessing these would be from Australian federal or Tasmanian government reporting on land assessments of threats and opportunities.

Some references have been added referring to the gap of current land monitoring system in Tasmania. Currently, Tasmanian scientists have developed an operational system to monitor

climate, and land suitability, but lack of maps for water content within soil, which is the main parameter for agricultural production.

Line 83. There has been little information share din intro about the nature of the Australia-wide data, or the nature of the modelling processes that would use these data. Deep learning is the modelling approach, there seems to be some associated of DL with SMAP, but there seems a gap in the narrative of how one goes from these features to using Australia-wide data to get daily soil moisture estimates. The model transfer bit seems well described, however. Not looking for detail here as this will come in the methods description, but the introduction seems to be a collection of loose ends and no clarity on what the intention of the work is.

Thanks for your comments. We realised the missing information about the nature of Australian moisture maps data in current intro section. One of the reasons is we were focused on replicating the work on downscaling SMAP using DL that has been done in Australia (Fuentes et al., 2022). Thus, we try to build story about why we chose the SMAP data. However, in the revised manuscript, we further discussed Australian data, and clarify the needs of regional SM maps in Tasmania.

Fuentes, I., Padarian, J., and Vervoort, R. W.: Towards near real-time national-scale soil water content monitoring using data fusion as a downscaling alternative, Journal of Hydrology, 609, 127705, https://doi.org/10.1016/j.jhydrol.2022.127705, 2022.

Line 84: "we contribute to" ?

This part has been revised accordingly.

Table 1. Given the public availability of daily gridded temperature and rainfall data from silo (https://www.longpaddock.qld.gov.au/silo/about/climate-variables/) for all of Australia (5km res) what would be the reason to go with ERA5 data?

As we are using cloud computing, we used the data that are available in Google Earth Engine database so that we could sample the data easier and quicker. This is interesting for the next, SILO dataset might better represent weather conditions across Australia continent, including Tasmania.

Table 1. The Searle et al 2022 reference pertains to just the AWC product.

Other references have been added.

Table 1. Australia also has a publicly available post processed version of the SRTM. Processed in terms of vegetation removal and hydro logically corrected (https://ecat.ga.gov.au/geonetwork/srv/eng/catalog.search#/metadata/69888). Would have though this were a better dataset to use.

This could be better option to use, thank you for your valuable suggestion. However, we have not considered it yet in this work since we are using the general information of topography as one of the covariates in our models. Topography (DEM) had lower importance in our models (Fig 13). Thus, we think the DEM SRTM that is available in GEE should be representative.

Table 1. For the land use lander mapping, am not sure this is the correct reference citation for this data source.

We have reviewed the reference citation and revised it to be more informative the land use mapper.

Table 2. Not clear about number of data points. What information is at these points?

This should be the number of SM data recorded from all stations, within the given period. We have revised this part.

Lines 120-149. These sections would be helpful if authors guided readers to fundamental research on these concepts. Maybe there could be some figures used in these sections to illustrate too. General feeling here is that this is all quite jargony.

We added some references for reader to look into for further details about each deep learning approaches that we used here.

Line 150. Maybe a map showing distribution of data and soil moisture sensors used both in across Australia and then Tasmania for focus.

The location of SM stations for Tasmania is presented in Fig. 1, while for Australia is in Fig. S1 supplementary text. We have added a sentence to specify this matter.

Line 153-54. Bit unclear about these data: "Reference soil moisture data were measured using frequency-domain reflectometry sensors available at different soil depths between stations". When was this collected, by whom and at what depths and frequency.

We have clarified this sentence and moved it to section 2.2 where we think more relevant.

Line 155. More detail needed about soil probe calibration. Saying based on bulk density information does not give enough info. Would be good to provide information about the type of sensors, general details.

More details on sensor device have been added in section 2.2, while the details about calibration SM data have been added in this section.

Line 163. What is meant by daily soil moisture was averaged? What part of the data is being averaged?

From the original data, we firstly calibrate the recorded moisture level by multiplying it with the total porosity at each station. After that, we aggregated the data from measurements at certain depth, into the mean measurements within a soil layer (using the spline method). Additionally, we aggregated the sub hourly data into daily average SM data.

This explanation has been added in the revised text.

Line 174. Explain the 'multiband image was calculated each day'

We mean that we have the multiband images ready for each date of prediction. We have revised this sentence accordingly.

Line 175-78. Need some clarity about these reference data. This was mentioned above too in my comments.

Reference data means the observed SM data for model development. We have clarified the part we mentioned as reference data.

Line 200-05. What soil moisture data is used in AU model?

For Australia, SM data were collected from the OzFlux and OzNet databases. The OzFlux provides SM data from flux monitoring tower set up to understand the exchanges of carbon and water between terrestrial ecosystems and atmosphere spreading across Australia. OzFlux stations use time-domain reflectometry Campbell Scientific sensor to record moisture level at specific soil depth that vary among the stations (see the details at https://www.ozflux.org.au/). Meanwhile, OzNet provides soil moisture records from several sites in the Murrumbidgee catchment, southern New South Wales, Australia. Each site measures soil moisture at at 0-5cm with soil dielectric sensor (Stevens Hydraprobe®) or 0-8cm, 0- 30cm, 30-60cm and 60-90cm with water content reflectometers (Campbell Scientific).

These details have been added to the revised document in section 2.2.

Line 207. Are each of these individual models combined into one?

No. We keep 39 models separately to calculate each model's performance.

Figure 2. So the out "Tasmania Soil Moisture Maps" is a combination of AU, Tas and transfer models, or is it the transfer model output as it was determined to be the best model?

The final maps were derived only from Transfer learning models with the LSTM algorithm as this combination showing the best performance.

Line 239. Are these the soil moisture sensor data and the 'other' data? This is not predicted or are they observed?

The SM data for models' development. We have clarified this sentence.

Figure 3. It is not clear what this data means or what can be interpreted from it. Just showing observed data all compiled together over the period of specified time does not provide anything too much informative.

Yes. We want to compare the distribution of SM data regardless the time and location of the data being recorded. This should inform the reader that we have pretty much similar distributions between Australia and Tasmania, thus when we can do the same modelling for both areas as we have almost similar target.

Line 260-67. So what does one make of these data. Pointing out some distinctness between data is fine, but what else are the authors trying to say here?

We want to show the nature difference of covariate values that we used for modelling in Australia and Tasmania. This likely makes the differences on the models' performance.

Figure 5. It would be useful to plot time series of soil moisture sensor data with SMAP to see consistencies of data through time.

We would like to have a comparable performance between the SMAP data, AU models, TAS models, and TL models towards the observed SM data. Since the prediction values (in scatter plots) were from multiple models' prediction resulted from Cross validation scheme, we think that would be too complicated to show them. Instead, we showed the time series plot for the final models that we chose for mapping.

Figure 6 and 7. What is shown as correlation is actually Lin's Concordance correction coefficient? Or is it actually just correlation?

It is just Pearson's correlation coefficient. We have specified that in the revised document.

Figure 9. Variation in RMSE seems higher compared with concordance. Even at top of Tasmanian, there is high concordance but some of those sites also have high RMSE. This is a bit of an odd outcome and should warrant an interpretation.

We have identified that this behaviour was coming from stations with small numbers of data, less than three months of SM data. Thus, we added a brief discussion regarding this matter.

Table 4. It is interesting that model outcomes are good for irrigated land use given model only considered rainfall and not any other supplementary inputs from irrigation. So model is adjusting for this given the data from the stations in this land use? This aspect seems to be overlook in discussion.

Yes, the models give a good performance for irrigation area, even we don't have irrigation data as input. We argue this might be the specialty of DL algorithm, which can adjust prediction value by merely being informed logical value of where the station is located. This could be an alternative for modelling over irrigated area with lack of irrigation records.

Figure 11. In western Tasmania, this are to my understanding has significant areas of shallow peats, that is peat thinly blanketing rock. Having estimates of SM for 30-60cm would therefore be unrealistic. Maybe this is affecting to model reliability? In any case, estimates of plant available water or simply hydrologically available water could be predicted quite a long way off from reality.

Yes. We also note this down as in our results showed strange predictions over this area (max moisture at surface layer, and min moisture at subsurface layer). We left the information as it is, just to show the end user that this area could be predicted, yet still unreliable at this point.

Line 413. "both were.." meaning the for the transfer modelling?

Yes, correct. We have clarified the sentence.

Line 414. Not sure i understand this thing about 'memory'. Maybe explain more. Other than the fact that model included variables to capture the latency between soil moisture and rainfall, what other memory is captured here?

It's just that, because we want to highlight why LSTM in AU models perform worse than that in TAS models. We have clarified this in the sentence.

Line 465-70. Soil thickness to consider here too

Yes. We have added a sentence regarding soil thickness in western part of Tasmania.

References

Frost, A.J., Ramchurn, A., Smith, A., 2016. The Bureau's Operational AWRA Landscape (AWRA-L) Model. Australian Bureau of Meteorology Technical Report.

Stenson, M., Searle, R., Malone, B. P., Sommer, A., Renzullo, L. J. and Di, H., 2021. Australia wide daily volumetric soil moisture estimates. Version 1.0 [Dataset]. Terrestrial Ecosystem Research Network, Canberra. https://doi.org/10.25901/b020-nm39.

Wimalathunge, N.S., Bishop, T.F.A., 2019. A space-time observation system for soil moisture in agricultural landscapes. Geoderma 344, 1-13.

**Reviewer 2**

The paper is well-written, although the data sets and the modelling approaches are not always clearly described. My main issue is the novelty and the real aim of the paper i.e. beyond a regional study.

The real aim of the paper should be reinforced in the introduction. As it stands the focus is on producing a soil moisture product for Tasmania, while the transfer learning is the scientific novelty. You mention in lines 70-76 that this technique has already been applied in other studies, but I cannot find any details on the extrapolation methods. Therefore, it is difficult for the reader to understand what the main challenges are that remain to be resolved are. What is the novelty of the use of transfer learning in Tasmania? The requirements for the soil moisture product are not clearly outlined and therefore there are no clear specifications of the soil moisture product. Unfortunately, these issues are not treated in the discussion section either.

Thank you for your comments.

We have revised the aim and novelty of our study: addressing the current data gap of daily high resolution soil moisture data by proposing scalable methods for soil moisture prediction in regions with limited observational infrastructure, thereby contributing to global efforts in sustainable land and water management. Specifically:

(i) a systematic evaluation of DL algorithms to identify the most effective approaches for downscaling SMAP datasets to finer spatial resolutions,

(ii) the innovative application of transfer learning in DL, utilising Australia-wide data to enhance soil moisture prediction accuracy in data-scarce regions like Tasmania,

(iii) comprehensive validation of the Tasmania-specific soil moisture map, providing a benchmark for future studies in areas with sparse observational data, and

The study further presents a demonstration of the feasibility of delivering live, daily soil moisture predictions, highlighting potential real-time applications in precision agriculture, water resource management, and environmental monitoring.

Line 78 Please provide some more details on the current land use change in Tasmania. It would also be good to explain the practical needs for soil moisture products that emanate from this land use change.

We have revised this part by adding the information of what's lack in current land monitoring system in Tasmania. This moisture maps extend information of moisture level recorded at some locations across Tasmania. This is the first attempt to generate high resolution maps only for Tasmania region based on this SM data records.

Table 1 The abbreviations in the 'variable/bands' column are not clear. What does 'band' refer to?

Band means the spatial data of the variable. We have clarified this column name into 'variable'.

Line 155 Why do you need bulk density data? The probes will already provide volumetric soil moisture content. Or is the recorded soil moisture data expressed as gravimetric moisture content. Please explain in some more detail.

The probes provide volumetric water level in percentage of pore space. They don't have specific calibration with soil properties. Thus, we used the bulk density data to account for soil porosity so that we can get proportion of water content within the whole soil body. We have clarified this in the revised text.

Line 171 How was the AWC calculated? Pedotransfer functions for AWC generally take SOC and clay content into account. Is not there a risk for redundancy?

According to Searle et al. (2022), the AWC maps was generated based on digital soil mapping approach using 154 individual raster data layers of environmental covariates in Australia. The covariate stack was used as the independent variable data for the predictions across all grid cells and at each depth. Thus, we think that these variables are not redundant.

Line 174 What does 'mulitband image' refer to? The collation of all co-variates or an image from a multispectral satellite?

This multiband image refers to the collation of all covariate's raster data. We have clarified this sentence.

Lines 203-204 The sentence is not entirely clear. Which variables were excluded? Why do you mention 'including' in the same sentence?

We excluded the mentioned variables: Sentinel-1 dataset, vegetation index, and land surface temperature. We have revised this sentence.

Line 213 Does the workflow of Fig. 2only describe the transfer learning in bullet point 'c' or all modelling scenarios? If the latter, please refer to Fig. 2 in line 200

The workflow is for all modelling scenarios. The text for referencing Fig.2 has been moved as suggested.

Line 224 For model evaluation the $R^2$ is generally used. Why do you prefer the 'r'?

We like to measure the direct relationship between predictions and observations. This could show how well predictions follow the observed patterns.

Figure 4 Please explain in the caption what the numbers refer to e.g. AWC1.

Thank you for noticing this. We have revised as suggested.

Line 270-275 I do not understand the difference between the two correlation approaches. Is the first approach for all stations and the second per probe location? Can you indicate the link to panels a and b in Fig 5?

Panel a shows evaluation of SMAP data at each probe location. Since we have 39 locations, we plot the evaluation results into a boxplot, to see how it distributes. Panel b shows the scatter plot of SMAP predictions and observations from all probe locations. We like to give the readers an extended view on how exactly the SMAP predictions behave on observations data.

Figs 11 and 12 When you discuss the model performance (e.g. Table 4), the standard deviation is based on the prediction in a number of stations (e.g. 3 for the forest). I am not clear what the standard deviation means in the maps (Figs 11 and 12. I might have missed it.

Standard deviation maps were derived from 39 map predictions resulted from cross validation schema. This represents the models' uncertainty in producing final maps. We have added a brief explanation about this in section 2.4.3.

Line 413 What does 'both' refer to?

It refers to both algorithms with TL approach. We have revised this part.

Line 454 Do I understand it correctly that the $R^2$ of predicted vs observed for soil moisture in some areas of China was 0.2? If so, could you please comment on the practical use of a prediction with such low $R^2$ values?

Low $R^2$ prediction could still provide valuable insights into trends, spatial patterns, or general relationships. This also could be a note for scientists to discover different models or scenarios that might predict better for the target variable.

Line 459 It is difficult to compare model performances based on the standard deviation. Uncertainty maps normally show the percentile interval range.

Thank you very much for your valuable insights. In this case, standard deviation should be able to inform readers about the overall variability of predictions from the 39 models. This also could give quick overview of which area has broader or narrower predictions variability.

Line 465 What does 'this area' refer to?

Western part of Tasmania. We have clarified this sentence.

Searle, R., Somarathna, P. D. S. N., and Malone, B.: Soil and Landscape Grid National Soil Attribute Maps - Available Volumetric Water Capacity (Percent) (3 arc second resolution) Version 2. v3. (v2), CSIRO [dataset], https://doi.org/10.25919/4jwj-na34, 2022.

---

## Author Response (AR3)

Dear Editors and Reviewer,

We thank you for your positive feedback. We have carefully addressed all minor technical corrections, including the writing issues and tense consistency noted by Reviewer #1. The final revised manuscript is now ready for production.

Please find the revised version attached. Kindly let us know if any further modifications are needed.

Thank you again for your support. We look forward to the next steps.

Best regards,
Marliana Tri Widyastuti (on behalf of all co-authors)